# Septin 9 Orients the Apico–Basal Polarity Axis and Controls Plasticity Signals

**DOI:** 10.3390/cells12141815

**Published:** 2023-07-09

**Authors:** Tingting Cai, Juan Peng, Mohyeddine Omrane, Nassima Benzoubir, Didier Samuel, Ama Gassama-Diagne

**Affiliations:** 1Unité 1193 INSERM, F-94800 Villejuif, France; tingtcai@foxmail.com (T.C.); nassima.benzoubir@inserm.fr (N.B.); didier.samuel@inserm.fr (D.S.); 2Université Paris-Saclay, UMR-S 1193, F-94800 Villejuif, France; 3AP-HP Hôpital Paul Brousse, Centre Hepato-Biliaire, F-94800 Villejuif, France

**Keywords:** septin 9, apico–basal polarity, polybasic domains, TGFβ/RhoA

## Abstract

The cytoskeleton is a master organizer of the cellular cortex and membrane trafficking and therefore plays a crucial role in apico–basal polarity. Septins form a family of GTPases that assemble into non-polar filaments, which bind to membranes and recruit cytoskeletal elements such as microtubules and actin using their polybasic (PB) domains, to perform their broad biological functions. Nevertheless, the role of septins and the significance of their membrane-binding ability in apico–basal polarity remains under-investigated. Here, using 3D cultures, we demonstrated that septin 9 localizes to the basolateral membrane (BM). Its depletion induces an inverted polarity phenotype, decreasing β-catenin at BM and increasing transforming growth factor (TGFβ) and Epithelial–Mesenchymal Transition (EMT) markers. Similar effects were observed after deleting its two PB domains. The mutant became cytoplasmic and apical. The cysts with an inverted polarity phenotype displayed an invasive phenotype, with src and cortactin accumulating at the peripheral membrane. The inhibition of TGFβ-receptor and RhoA rescued the polarized phenotype, although the cysts from overexpressed septin 9 overgrew and presented a filled lumen. Both phenotypes corresponded to tumor features. This suggests that septin 9 expression, along with its assembly through the two PB domains, is essential for establishing and maintaining apico–basal polarity against tumor development.

## 1. Introduction

Many organs are composed of epithelial cells polarized along their apico–basal axis. The apical membrane faces the lumen of a tube or the external environment, and the basolateral membrane (BM) binds to neighboring cells and the basal extracellular matrix (ECM). The generation of this polarized architecture and lumen formation is critical to the correct development of many organs and to adult mammalian tissue homeostasis [1]. A loss of polarity and orientation with respect to the apico–basal axis, which usually occurs early in tumorigenesis, leads to changes in plasticity and the Epithelial–Mesenchymal Transition (EMT). However, he molecular mechanisms that control polarity and EMT require further investigation [2,3].

Cytoskeleton-dependent membrane shaping processes control organelle positioning and endosomal sorting to the specific membrane domains that drive apico–basal polarity and lumen formation in epithelia [4,5]. The polarization process of epithelial cells is often accompanied by the formation of primary cilia protruding into the lumen, which is controlled by polarity components [6,7,8]. The roles of microtubules and actin in these dynamic processes have been described in detail, although information about the roles of other cytoskeletal elements [9], such as septins, is only just emerging [10,11]. Septins are guanosine triphosphate (GTP)-binding proteins that are evolutionally conserved among all eukaryotes except plants and certain protists [12,13]. Septins are involved in infection by pathogens, including bacteria, fungi, and viruses [9,14,15]. The deregulation of septin expression has been linked to severe diseases, including cancers and neurological diseases [16,17,18]. 

The septin family comprises 13 members in mammals, and the complexity of this gene family is increased by the existence of alternate splicing, which dramatically increases the number of isoforms [17]. Septins form hetero-oligomeric complexes through binding to the membrane and may develop high-order structures such as filaments and rings. These structures act as a diffusion barrier and play a scaffold role in different cellular processes by recruiting cytosolic proteins and cytoskeletal elements such as microtubules or actin during membrane remodeling processes [16,17]. Septin 9 regulates cell polarity and cilium organization and assembly [19]. The role of septins in cell–cell junctions has also been reported [20,21]. However, the effects of septins, particularly septin 9, on epithelial cell morphogenesis and plasticity, lumen and cilia formation, and ECM assembly were not investigated during the same study.

The association of septins to the membrane is mediated by their binding to PIs via a characterized PB1 domain [14,22,23]. We recently identified a second polybasic domain in septin 9 (PB2) within the GTP-binding domain, which is conserved among the septin family. Furthermore, we showed that both PB1 and PB2 are required for septin 9 assembly in filaments with septin 2, 6, 7 and for the assembly and functionality of the Golgi apparatus [24]. Therefore, this suggests a crucial role for these PB domains in septin function and apico–basal polarity.

For the present study, we investigated the role of septin 9 in the establishment of apico–basal polarity and EMT using MDCK (Madin Darby Canine Kidney) cells grown under 2D conditions on Transwell filters and in a 3D culture in Matrigel. We also assessed the mechanisms involved and the contribution of PB domains. 

## 2. Materials and Methods

### 2.1. Cell Lines and Culture Conditions 

MDCK (Madin Darby Canine Kidney) cells were maintained in minimal essential medium (MEM; Invitrogen, France) containing Earle’s balanced salt solution supplemented with 5% fetal bovine serum and 1% Penicillin/Streptomycin solution. For the 2D culture, about 200,000 MDCK cells were plated on 0.4-µm polycarbonate Transwell Filters (12 mm diameter, Corning, France) for 3 days to polarize. For 3D cultures, MDCK cells were trypsinized as 10,000 single cells/mL in 2% Matrigel (BD Biosciences, France); 500 μL of cells were plated in each well of 8-well Lab-Tek II chamber slides (Thermo Fisher Scientific, France) pre-coated with Matrigel and grown for 1, 2, 4 or 6 days as the cysts with lumens formed, as previously described. For transfection, MDCK cells were seeded at density (2.5–3 × 10^4^ cells/cm^2^) in 12-well cell culture microplates and then transfected with 1 µg cDNA or 110 pmol siRNA per well using jet PRIMETM (Ozyme, France) according to the manufacturer’s instructions. Regarding 3D culture, 24 h after transfection, the cells were detached with trypsin and plated on Matrigel for 4 days, as indicated above. 

The specific septin 9 duplex small-interfering RNAs (siRNAs) used came from Thermo Fisher, and the scrambled siRNA (control) came from Santa Cruz. The sequences are presented below:

si Septin 9 (SEPT9HSS173896) (si2): 5′-AGGCGCCUGCAUCACGGAACGAGAA-3′, 5′-UUCUCGUUCCGUGAUGCAGGCGCCU-3′.

si Septin 9 (SEPT9HSS173897) (si3): 5′-GCCAUGAAGCAGGGCUUCGAGUUCA-3′, 5′- UGAACUCGAAGCCCUGCUUCAUGGC-3′.

### 2.2. Antibodies 

Anti-Septin 9 Cat#ab114099 (WB:1/500, IF:1/25), anti-septin 2 Cat# ab88657 (IF:1/100), mouse and rabbit anti-V5 tag Cat#ab27671 (WB:1/1000, IF:1/400), and Cat#ab9116 (WB:1/1000, IF:1/400) were obtained from Abcam; anti β-tubulin Cat# T4026 (IF:1/100) was purchased from Sigma-Aldrich; anti-E-cadherin Cat#610181 (WB:1/1000, IF:1/100) and anti-GM130 Cat#610822 (IF:1/50) were sourced from BD Biosciences. Anti-Na^+^/K^+^ATPase Cat#33-9100 (WB:1/1000, IF:1/100) and ZO-1 Cat#33-9100 (WB:1/1000, IF:1/100) came from Invitrogen. Anti-cortactin Cat#3503 (WB:1/1000, IF:1/100), anti-p-cortactin Cat#4569s (WB:1/1000, IF:1/100), anti-src Cat#2109s (WB:1/1000, IF:1/100), and anti-p-src Cat#2101s (WB:1/1000, IF:1/100) were purchased from Cell Signaling Technology. Anti-septin 7 Cat# sc-20620 (IF:1/100), anti-β-catenin sc-7199 (IF:1/100), anti-actin Cat#sc-1616 (WB:1/1000), anti-fibronectin Cat#sc-6952 (WB:1/1000), and anti-Integrin β1 Cat#sc-18887 (WB:1/1000) were obtained from Santa Cruz. The GP135 antibody (1:1000) came from George Ojakian (University of New York Downstate Medical Center), USA. Anti-laminin Cat# L9393 (WB:1/1000) was purchased from Sigma. Anti-Vimentin Cat# CBL202 (WB:1/1000) was obtained from Millipore. Alexa Fluor™ 568 Phalloidin Cat#A12380 (IF:1/100) and Hoechst Cat#34580 (1:10000) were purchased from Life Technology. Secondary antibodies: Alexa Fluor^®^ 633-, 546-, and -488-conjugated were purchased from Life Technology; Hoechst was purchased from Molecular Probes.

### 2.3. cDNA Constructs and Stable Cell Lines 

The pcDNA3.1/V5-His-TOPO vector containing either the cDNA of the septin 9 isoform 1 (septin 9_i1) or the cDNA of the mutants of the septin 9 isoform 1deleted from PB1 domain (septin 9_del1), PB2 domain (septin 9_del2), or PB1 and PB2 (septin 9_del1,2) (Appendix A) were constructed using the QuikChange II XL Site-Directed Mutagenesis Kit (Cat#200521) as previously reported [14,24] and used to stably transfect the MDCK cells. The cells were then treated with 700μg/mL neomycin (Invitrogen) for 2 weeks. Stably transfected pools were seeded at 0.5–1 cell/well in 96-well plates to limit dilution. After another 3–4 weeks, the colonies were generated (Appendix A) and subsequently analyzed using Immunoblot and Immuno-fluorescence. Only the colonies that were positive when assessed using both methods were selected for further investigation.

### 2.4. Immunofluorescence Staining 

Cells were grown on coverslips, fixed for 15 min with paraformaldehyde 4%, and permeabilized for 30 min with PFS Buffer (PBS+ containing 0.025% m/v saponin and 0.7% gelatin) at 37 °C. The cells were then incubated with in primary antibodies for 2 h and with appropriate secondary antibodies for 90 min. The coverslips were mounted using Prolong Gold (Invitrogen). For the 3D culture, the samples were incubated in primary antibodies at 4 °C overnight followed by incubation for 90 min at 37 °C, and then treated as described for the 2D culture. 

### 2.5. Image Acquisition and Analysis 

Images were acquired with a Leica TCS SP5 AOBS tandem confocal microscope. The 3D images included in this paper were obtained using Icy bioimage analysis 2.3.0.0 software (http://icy.bioimageanalysis.org, accessed on 01 June 2014. To quantify the intensities and distribution of the fluorescence signals, confocal Z stack images were processed using the Image J background subtraction tool.

### 2.6. Immunoblot 

The cells were washed with ice-cold Dulbecco’s Phosphate Buffered Saline (DPBS) and lysed on ice in the following buffer: 20 mM Tris, HCl, 100 mM NaCl, 1% Triton X100, and 10 mM EDTA at PH 7.4 containing a protease and phosphatase inhibitor cocktail (Complete™ ULTRA Cat#05892970001 Roche). The proteins were separated on SDS (sodium dodecyl sulfate) polyacrylamide gel and electro-transferred onto nitrocellulose membranes. After transfer, the membranes were saturated in DPBS containing 0.1% Tween 20 and 5% milk. Primary antibodies were added overnight at 4 °C or for 2 h at room temperature depending on the antibody. The membranes were then washed with DPBS and incubated for 1 h at room temperature with appropriate secondary antibodies coupled with peroxidase. The ECL plus kit (Cat#32132) and SuperSignal™ West Femto Maximum Sensitivity Substrate (Cat#34095), obtained from Thermo Scientific, were used for protein detection. Chemiluminescent signals were detected by the G: BOX Chemi Fluorescent & Chemiluminescent Imaging System from SYNGENE. The blots were quantified using Image J 1.53t.

### 2.7. Cell fractionation Assay

The confluent monolayers of cells were placed on ice and washed twice with ice-cold PBS at pH 7.4 before the addition of 10 mM Tris/HCl (pH 7.4) buffer for 1 min. The cells were scraped into a homogenization buffer comprising 10 mM Tris/HCl, 1 mM EGTA, 0.5 mM EDTA, and 0.25 M sucrose at pH 7.4, which also contained complete^TM^ protease inhibitors. All centrifugation processes were performed at 4 °C, and the samples were kept on ice throughout the procedure. The combination was centrifuged first at 720× *g* for 5 min. The supernatant was then transferred into a fresh tube and ultracentrifuged at 100,000× *g* for 1 h (the supernatant being the cytoplasm and the pellet the membrane). The solution was then resuspended in an appropriate TBS with 0.1% SDS for immunoblot.

### 2.8. Quantitative Reverse Transcription-PCR (qRT-PCR) Assay

Total RNA was isolated using a RNeasy Mini Kit 50 (Qiagen). Reverse transcription was performed using the Reverted First Stand cDNA Synthesis Kit (Fermentas). cDNA was amplified using the Quanti Tect SYBR Green PCR Kit (QIAGEN) and the 7500 Fast Real-Time PCR System (Applied Biosystems). The reaction program was 95 °C for 10 min, followed by 40 cycles of 95 °C for 15 s, 55 °C for 30 s, and 72 °C for 30 s. The mRNA level was normalized to GAPDH expression. The primers are listed in Appendix A.

### 2.9. Statistical Analysis 

The statistical significance of the immunofluorescence, immunoblotting, and RT-qPCR findings were determined via Student’s *t*-test using Microsoft Office Excel 2019 software (Microsoft Corporation). The results showed the means and standard deviations, and those with *p* values lower than 0.05 were considered to be statistically significant (* *p* < 0.05, ** *p* < 0.01, ****p* < 0.001). All tests were two-tailed.

## 3. Results

### 3.1. Septin 9 Is Essential to Orient the Apico–Basal Axis, and Its Knock Down Inverts Polarity

To decipher the role that septin 9 might play in the establishment of apico–basal polarity, we used MDCK cells cultured in 3D in Matrigel as a model system. After 4 days in culture, the control cells formed cysts with a monolayer of polarized cells surrounding an open central lumen. Endogenous septin 9 was present at the BM in control cells (Figure 1A), similarly to septin 9 expression in the kidney section (Appendix A). Then, two different siRNA were used to knock down septin 9 (Appendix A), and treatment with siSeptin 9-3 (siSeptin 9) led to the formation of a high proportion of cysts with multiple lumens and inverted polarity (Figure 1A,B). The assembly of septin 9 in filaments is required to perform its biological activities. The filaments are initially formed by an octameric complex that includes septin 2, septin 6, and septin 7 [25] (Appendix A). Therefore, we analyzed septin 7 and septin 2 and found that the two proteins were located at the BM, as observed for septin 9, and that the depletion of septin 9 decreased their expression, as shown via immunofluorescence analysis (Appendix A) and Western blot analysis (Appendix A). These data were in line with our previous report [14] and that of others, indicating that the treatment of cells with septin 7 siRNA leads to a reduction in the levels of other septins involved in the heteromeric structure [26,27]. Taken together, these data strongly suggest that septin 9 is integrated in filament structures at the BM and that its expression is required for the stabilization of the complex formed with septin 2 and septin 7.

The structure, organization, and positioning of the Golgi apparatus are implicated in the apico–basal polarization process [28]. Given that septin 9 regulates the asymmetric assembly of the Golgi apparatus [24], we assessed whether the disrupted apical–basal polarization seen in septin 9-knockdown cysts also affected Golgi organization. Therefore, we stained for Golgi using an antibody against the Golgi matrix protein GM130. In control cells, the Golgi was localized as expected above the nucleus in the apical part of the cytoplasm toward the cyst lumen (Figure 1C). By contrast, in the inverted polarized cysts formed by cells treated with siRNA septin 9, the Golgi was disorganized and localized behind the nucleus in the basal part of the cytoplasm beneath the intense actin signal at the cell periphery (Figure 1C–F). Taken together, these data suggest a specific role for septin 9 in the Golgi positioning and orientation of the apico–basal polarity axis.

### 3.2. Septin 9 Regulates Cell–Cell Junctions and BM Stability 

Both the correct localization and assembly of cell–cell junctions are essential to establishing apico–basal polarity and BM stability (Figure 2A). At the BM, β-catenin links the cytoplasmic domain of E-cadherin to actin-binding proteins such as cortactin to organize the cortical actin networks and thus maintain the adherens junctions [29] (Figure 2A). In light of the localization of septin 9 at the BM membrane and its effects on apico–basal polarity, we investigated the consequences of septin 9 depletion on cell–cell junctions. We also analyzed the tight junction protein Zonula occludens-1 (ZO-1), which plays a crucial role in the conversion of AJs to belt-like structures in polarized epithelial cells [30]. The results showed that the knock-down of septin 9 expression using siRNA decreased that of both E-cadherin and β-catenin (Figure 2B). A loss of cortactin due to septin 9 depletion was also verified by immunofluorescence analysis in a 3D culture (Figure 2B). These immunofluorescence data were validated by immunoblots (Figure 2C).

The loss of cell–cell junctions is a fundamental event during EMT, a process by which epithelial cells lose polarity to acquire the mesenchymal and invasive features that characterize cancer cells [31]. Using qRT-PCR analysis, we were able to show that a decrease in septin 9 significantly increased the gene expression of mesenchymal cell markers such as vimentin, EMT-inducing transcriptional factor (ZEB1), and N-cadherin (Figure 2D). Overall, these results suggest a potential function for septins in driving the assembly, maintenance, and remodeling of adhesion junctions, preventing cells from undergoing EMT. Thus, septin 9 regulates cell–cell junctions at BM and appears to act as a gatekeeper against EMT.

### 3.3. The Two PB Domains of Septin 9 Are Required for Its Basolateral Localization and Apico–Basal Polarity

We have reported that septins have a second polybasic domain (PB2) that with PB1 forms a basic cluster at the NC interface. In particular, we found that septin 9 PB domains control the formation and assembly of its filamentous structure and the functionality of the Golgi apparatus [24](Appendix A). Accordingly, we tried to determine whether both PB domains of septin 9 were required for its role in apico–basal polarity. Thus, we established different MDCK cells that stably expressed either empty vector (EV), the V5 tagged isoform 1 of septin 9 (septin 9_i1), mutants of septin 9_i1 deleted from either the PB1 domain (septin 9_del1) or the PB2 domain (septin 9_del2), or mutants deleted from both the PB1 and PB2 domains (septin 9_del1.2) (Appendix A). The expression of the septin 9 protein in these cells was validated first by immunoblot using the V5 tag antibody (Figure 3A). All these different cell lines were then cultured in 3D to form cysts and stained for V5 tag and actin (Figure 3B). The EV- and septin 9_i1-expressing cells formed cysts with an open central lumen, and septin 9_i1 was enriched at the BM, as shown for endogenous septin 9 (Figure 1A). However, the lumen was more likely to be filled with septin 9_i1 cells rather than EV cells (Figure 3B). By contrast, the septin 9_del1 and del2 mutants formed multi-lumen cysts, while the septin 9_del1.2 cells mostly formed cysts with an inverted phenotype (Figure 3B). The quantification of these different phenotypes is presented in Figure 3C. Importantly, we observed the cytoplasmic localization of septin 9_del1, septin 9_del2 and septin 9_del1.2 (Figure 3B). The increased septin 9 del1.2 cytoplasm accumulation was confirmed by cell fractionation (Figure 3D), thus confirming that PB domains are required for septin 9 assembly [24]. Taken together, these data indicate that the PB domains of septin 9 regulated its basolateral localization at the cell cortex, affecting cell shape and single lumen formation. We also validated the finding that the Golgi was localized above the nucleus facing the lumen in EV and septin_i1 cysts, whereas in septin 9_del1.2 cells which formed cysts with an inverted phenotype, the Golgi was localized at the cell periphery (Figure 3E–G). Furthermore, the deletion of the two PB domains induced the presence of ZO-1 at the periphery of del1.2 cysts with the inverted polarity phenotype (Appendix A). In summary, preventing the assembly of septin 9 by deleting its two PB domains inverted apico–basal polarity.

### 3.4. The Two PB Domains of Septin 9 Are Critical to Maintaining Cell–Cell Junctions and BM Stability

In light of the data presented above (Figure 2B,C), we explored the effects of deleting the two PB domains on E-cadherin expression (Figure 4A–C). We did not observe a striking decrease in E-cadherin expression; instead, E-cadherin decreased from the membrane and became more cytoplasmic, as highlighted by the line plots shown in Figure 4B,C. Furthermore, the cells were grown in 2D on semi-permeable filters. Even though this system did not allow for lumen formation, it enabled a different analysis of the apico–basal axis. After 3 days of culture, the EV cells formed a monolayer, as seen from the X-Y and X-Z sections after the staining of β-catenin (Figure 4D). In cells expressing septin 9_i1, β-catenin was well organized, and septin 9 was enriched in the BM (clearly visible in the X-Z section) and formed filaments, mostly at the basal part (Figure 4D). By contrast, septin 9_del1.2 lost its filamentous structure and was delocalized on the apical domain, as better seen in the X-Z section (Figure 4D). Remarkably, in septin 9_del1.2 cysts, the β-catenin signal was diffused and markedly reduced at the zonula adherens (ZAs), as indicated by the line plot and the 3D reconstruction of the septin 9 and β-catenin signals in a single cell from each group (Figure 4E,F). These results were supported by quantification of the β-catenin cellular signal (Figure 4G). Overall, the data demonstrate that the two PB domains of septin 9 are required for its filaments and basolateral localization and for maintaining cell–cell adhesion.

### 3.5. The Two PB Domains of Septin 9 Regulate Cell–ECM Adhesion

To further characterize the inverted polarized cysts formed by del1.2 cells, we stained Na^+^/K^+^ ATPase, a key component in the maintenance of the epithelial phenotype that is present at the BM of most epithelial cells [32]. Indeed, Na^+^/K^+^ ATPase was found at the BM of the EV cells, and the signal was stronger in i1 cells. Surprisingly, Na^+^/K^+^ ATPase was absent from del1.2 cells, as evidenced by the 3D reconstruction of the confocal data shown in Appendix A. These data echoed that of our previous report on the absence of integrins and ECM components such as laminin and collagen IV from cysts with an inverted polarity phenotype [33]. Interestingly, Integrin β1 decreased in laminin septin 9_del1.2 cysts compared to septin 9_i1 cysts (Appendix A), and laminin was absent from del1,2 cysts (Appendix A). Furthermore, we tested fibronectin, an important multifunctional protein in the ECM [34,35,36]. Fibronectin increased in septin 9_il cysts compared to EV cysts and formed well-organized structures in the center of the cysts (Appendix A), while this had totally disappeared from the septin 9_del1.2 cysts. However, the signal remained at the periphery of septin 9_del1.2 cysts and was stronger than that of EV cysts (Appendix A). Therefore, we concluded that deleting the two PB domains of septin 9_i1 impaired the ECM assembly and cell–ECM adhesion required to control apico–basal polarity.

### 3.6. The PB Domains of Septin 9 Regulate Lumen Formation at Different Stages of the Polarization Process

To further decipher the importance of PB domains during the polarization process, the different MDCK cell lines were cultured under 3D conditions and analyzed for β-catenin and the apical marker GP135 at different time points, starting from apical membrane initiation site (AMIS) formation after 1 day, the preapical patch (PAP) after 2 days, and open lumen formation after 4 days [37], and then followed the cysts cultured for up to 6 days. In EV and septin 9_i1cysts, the AMIS was clearly visible and it was apparent that the PAP had formed, while the lumen continued to grow until day 6 (Figure 5A). Here, again, and as observed in Figure 3C, the lumen of septin 9_i1 cysts started to fill, which suggested an overgrowth of the cells. However, no AMIS was present in the mutant cells, and multiple lumens were already present at the PAP stage. At day 4 and day 6, both del1 and del2 cysts displayed the multiple lumen phenotype. Del1.2 cysts presented the inverted polarity phenotype, as expected (Figure 5A); this phenotype was exacerbated at day 6, and the cysts displayed invasive features differing from the round shape of the cysts at day 4. The apical marker presented an asymmetric localization that recalled that of the front-rear polarization in migrating cells [38] (Figure 5A). The cysts forming AMIS (Figure 5B,C) and those with the different phenotype were quantified under each condition and at different time points (Figure 5D). A schematic representation of this study is shown in Figure 5E.

### 3.7. Deletion of the Two PB Domains of Septin 9 Promotes Invasive Features through the Regulation of Src and Cortactin

To obtain more mechanistic insights into how the deletion of the two PB domains of septin 9 induced the invasive phenotype, we studied the cysts formed from EV, septin 9_i1, and septin 9_del1.2 after 6 days of 3D culture. We found that septin 9 was cytoplasmic on day 6 in each cell cyst (Appendix A), similar to what was seen on day 4 (Figure 3C). Interestingly, septin 9 was enriched in the front of cysts with the invasive phenotype (Appendix A). We analyzed cortactin which was distributed uniformly on the BM membrane and particularly on the basal part in EV cysts. A similar staining was observed for septin 9_il cysts. By contrast, the cortactin signal increased markedly and colocalized with actin in septin 9_del1.2 cysts (Figure 6A). Furthermore, the phosphorylation of cortactin (p-cortactin), reflecting its activity, displayed a much more striking signal in septin 9_del1.2 cells (Figure 6A). As cortactin is the substrate of src [39,40,41] and the src phosphorylation of cortactin enhances actin assembly [42], we analyzed src. Interestingly, we found that src and p-src expression increased in septin 9_del1.2 cysts (Figure 6B), a finding confirmed by immunoblot (Figure 6C and Appendix A). Thus, these data indicate that deleting two PB domains of septin 9 induces the activation and recruitment of src/cortactin, allowing the cells to acquire invasive features.

### 3.8. Inhibition of the RhoA and TGF-β Type I Receptor Rescues the Polarity of del1,2 Cysts

RhoA is the first family member of the Rho family of GTPases, which control the actin dynamic, and is regulated by cortactin [34,43]. Furthermore, the activation of RhoA in cysts with an inverted polarity phenotype was previously reported, and the inhibition of Rho-associated kinase (ROCK) prevented this inversion [38,44]. Therefore, we wanted to determine whether the inverted polarity and invasive phenotype of septin 9_del1.2 cysts were dependent on RhoA. We then examined the effects of the ROCK inhibitor (Y27632) on both control (EV) and septin 9_del1.2 cells (Del1,2) (Appendix A and Figure 7A). Interestingly, treating del1.2 cells with Y27632 at 30 and 50 μM reverted the polarity, and the apical domain was found in the interior as either single or multiple lumens (Figure 7A,B) and accounted for 47.7% and 61% of the cysts at Y27632 doses of 30 and 50 μM, respectively (Figure 7A,B). We then assessed RhoA activity, which, as expected, increased significantly in septin 9_del1.2 cells compared to EV cells (Figure 7C). Therefore, these findings indicated that RhoA-dependent-actomyosin contractility is involved in the formation of the inverted polarity phenotype of del1,2 cysts.

Transforming growth factor-β (TGF-β) is a master regulator of epithelial cell plasticity and a driver of EMT [31]. Furthermore, the TGF-β mediated activation of RhoA maintains basal RhoA–ROCK signaling [44,45]. Therefore, we analyzed TGF-β signaling in cysts resulting from the depletion of the two PB domains of septin 9 with an invasive phenotype. First, we treated the cells with TGF-β type 1 receptor inhibitor (SB431542) at two different concentrations (Figure 7D). Strikingly, this treatment hampered the inversion of polarity and allowed for the formation of a single monolayer of cells and the recovery of β-catenin at cell–cell junctions (Figure 7D, E). Subsequently, we showed a decrease in E-cadherin and an increase in vimentin via performing immunoblot on EV and del1,2 cells (Figure 7F). Next, using qRT-PCR, we showed an increase in the transcripts of other mesenchymal markers such as ZEB1 and N-Cadherin (Figure 7G). Interestingly, we also observed an increase in TGF-β (Figure 7G). Together, these data reveal the role of TGF-β-driving plasticity regarding the invasiveness feature induced by depleting the two PB domains of septin 9. Therefore, they strongly suggest involvement of the EMT process regulated by the TGFβ/RhoA/cortactin pathway (Figure 7H).

## 4. Discussion

In this study, we have shown that septin 9 is required to orient apico–basal polarity axis and described how this function is controlled by its two PB domains, which are required for septin 9 assembly.

Septins are considered as a fourth cytoskeletal element, and most of the multiple biological functions they perform are dependent on their membrane binding and filamentous structure formation [9]. Despite evidence regarding the crucial role played by PB domains in septin filament formation and their binding to phospholipids on membranes, their role in septin functions remains under-investigated. In particular, septin 9 forms a filament structure by assembling with septin 2, septin 6, and septin 7 [25]. Interestingly, during our study, we found that septin 2 and septin 7 were localized at the BM in polarized MDCK cysts (Appendix A) and that their expressions were decreased by the knock-down of septin 9 using siRNA. The cell–cell junctions at the BM, including adherens junctions and tight junctions, are required for the polarization of epithelial cells and integrity of the epithelium. Interestingly, we also found that septin 9 regulated the expression of both E-cadherin and β-catenin at the BM, as well as that of the tight junction protein ZO-1. These findings were also supported by those of two other recent works [20,21]. One of these works reported that septin 2 was localized at cell–cell junctions in human microvascular endothelial monolayers and that this is necessary for the organization of cell–cell adhesion [20]. The other showed that the septin complex, containing septin 2, septin 6, and septin 7, acts as a scaffold to recruit β-catenin to synergize with E-cadherin, thus linking the catenin complex to the actin cytoskeleton in order to establish epithelial cell polarity [21]. Furthermore, the deletion or disruption of septins liberated the adherens junctions and polarity components to the cytoplasm, inducing deformation of the apical lumen in cysts under 3D conditions [21].

Importantly, we found that the depletion of septin 9 or deletion of the two PB domains induced the formation of cysts with an inverted polarity phenotype. Interestingly, we demonstrated that the septin 9 mutants with deleted PB domains then accumulated in the cytoplasm above the nuclei in the apical region. However, the overexpression of full-length septin 9_i1 increased cell height. We then proposed that septin 9 assembled with septin 2 and septin 7 to form filaments acting as a scaffold to link the plasma membrane to the actin cytoskeleton through the two PB domains and thus stabilized adherens junctions to maintain apico–basal polarity. The inverted polarity phenotype has been linked to collective cell migration and invasion [46]. Collective migration behavior has also been associated with the disruption of cell–cell junctions, allowing for protrusion in vivo and in vitro [47,48]. Furthermore, we found that cortactin and src were activated in the inverted phenotype cysts with septin 9_del1.2 expression. Cortactin is a regulator of actin polymerization at the membrane cortex and regulates late endosome trafficking by inducing branched actin assembly, which subsequently impacts Golgi homeostasis [49]. Cortactin is required for the assembly of protrusion in cancer cells and is important for cell migration and invasion [50,51,52]. Furthermore, cortactin has been identified as a target of E-cadherin-activated Src family kinase signaling to support cadherin adhesion and the integrity of cell–cell contact [53]. Moreover, it has been shown that septin 6 cooperates with septin 7 to coordinate actin remodeling and microtubules to promote the formation of filopodia by increasing cortactin recruitment [39]. Thus, the activation of cortactin and src in the inverted polarized cysts revealed that the balance of septin 9 expression was required to maintain epithelium homeostasis. Indeed, overexpression induced the proliferative tumor phenotype with the lumen filling of MDCK cysts, and the loss of expression or deletion of two septin 9 PB domains induced the invasive tumor phenotype represented by the inverted polarized phenotype. Overall, our results demonstrate a central role for the newly identified PB2 domain [24] in the biological function of septin 9. Thus, septin 9 could be seen as the central regulator of the apico–basal polarity of epithelial cells and a guardian against cancer cell migration and invasion, thus remaining an important target in the study of aggressive carcinomas.

## Figures and Tables

**Figure 1 cells-12-01815-f001:**
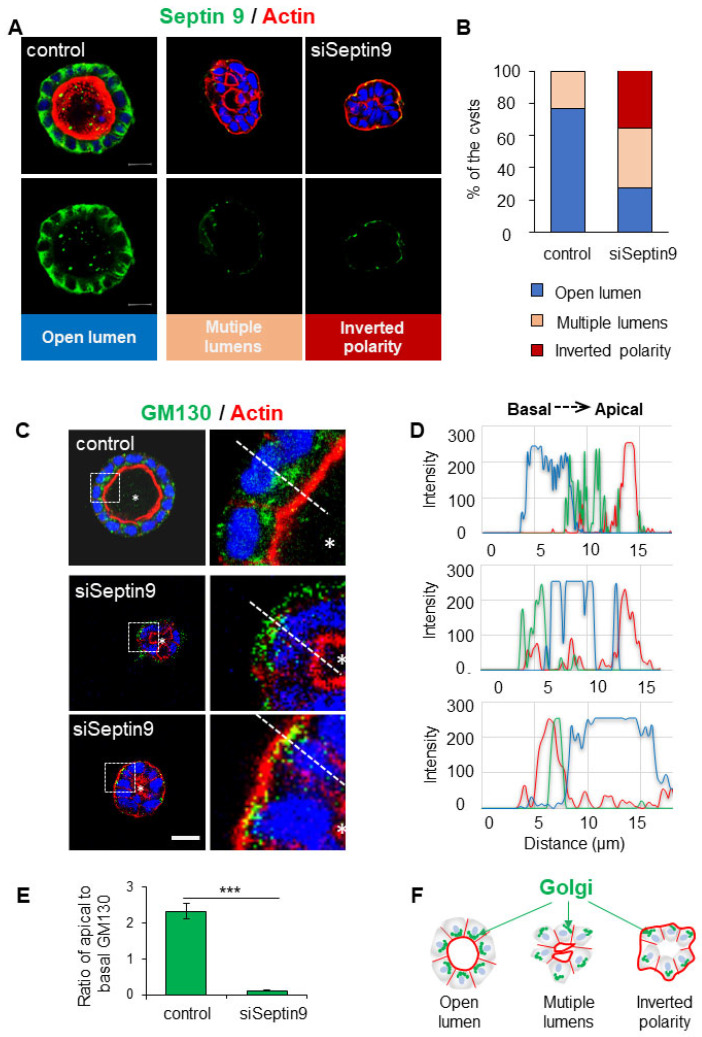
Septin 9 localized in the basolateral membrane is required for the orientation of the apico–basal axis in MDCK cysts. (**A**) MDCK cells were transfected with scrambled siRNA (control) or with siRNA of septin 9 for 24 h and then transferred to the Matrigel-covered Lab-Tek. The cells were incubated for 4 days to form cysts. Cells were stained for Septin 9 (green), Actin (red), and Hoechst (blue). A single confocal section through the middle of a cyst is shown. Scale bar 10 µm. (**B**) Cysts presented different phenotypes, including a single open lumen, multiple lumens, and inverted polarity. The color for each phenotype in a column corresponds to the color below the images. (**C**) Another experiment; as in (**A**), MDCK cysts sections labeled for the Golgi marker GM130 (green), actin (red), and Hoechst (blue). Higher magnifications of the boxed areas are shown. (**D**) Line profiles showing Golgi distribution from the apical to the basal areas, obtained using Image J. (**E**) Quantification of Golgi distribution: the ratio of maximal fluorescence intensity at the apical side versus the maximal fluorescence intensity at the basal side. (**F**) The schematic graph shows the Golgi distributions associated with different cyst phenotypes including cell polarity and lumen formation. Data information: Data concern at least two replicates. *n* = 69 (Control); *n* = 32 (siRNA) (**B**). *n* = 10 (**E**). The statistical values are means ± s.e.m. Student’s *t*-test was used. *** *p*  <  0.001.

**Figure 2 cells-12-01815-f002:**
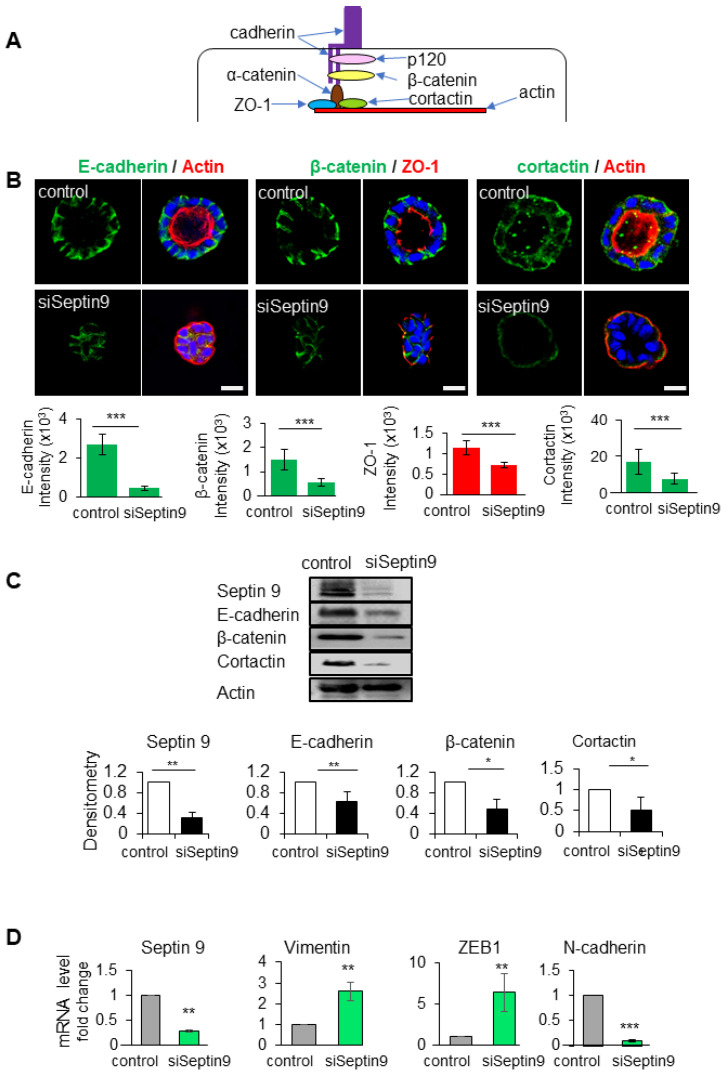
Septin 9 regulates and impacts cell–cell adhesion and BM stability. (**A**) Schematic representation of the cell–cell adhesion structure. In the cytoplasm, the E-cadherin/catenin complex links to ZO-1, actin, and the actin-binding protein cortactin. (**B**) MDCK cells were transfected with scrambled siRNA (control) or with septin 9 siRNA for 24 h and plated on Matrigel for 4 days to form cysts and then stained for E-cadherin (green) and actin (red), β-catenin (green) and ZO-1 (red), and cortactin (green) and actin (red). A single confocal section through the middle of a cyst is shown. Quantification of the fluorescence intensity of each protein expression: E-cadherin (green), β-catenin (green), and ZO-1 (red). Scale bar 10 µm. (**C**) MDCK cells were transfected with scrambled siRNA (control) or with septin 9 siRNA (siSeptin9) for 24 h, then the medium was changed to continue grown on plates for 3 days. Cells were lysed and analyzed by Western blot to determine the expression of septin 9, E-cadherin, β-catenin, and cortactin proteins. (**D**) Another experiment; as in (**C**), qRT-PCR analysis of the expression of mRNA encoding septin 9, vimentin, N-cadherin, and ZEB1 in cells with control and septin 9 knock-down under 2D conditions. Data information: Data concern at least two replicates and cysts (*n* > 10) for 3D staining, three replicates for immunoblot and qRT-PCR. The statistical values are means ± s.e.m. Student’s *t*-test was used. * *p* < 0.05, ** *p* < 0.01, *** *p*  <  0.001.

**Figure 3 cells-12-01815-f003:**
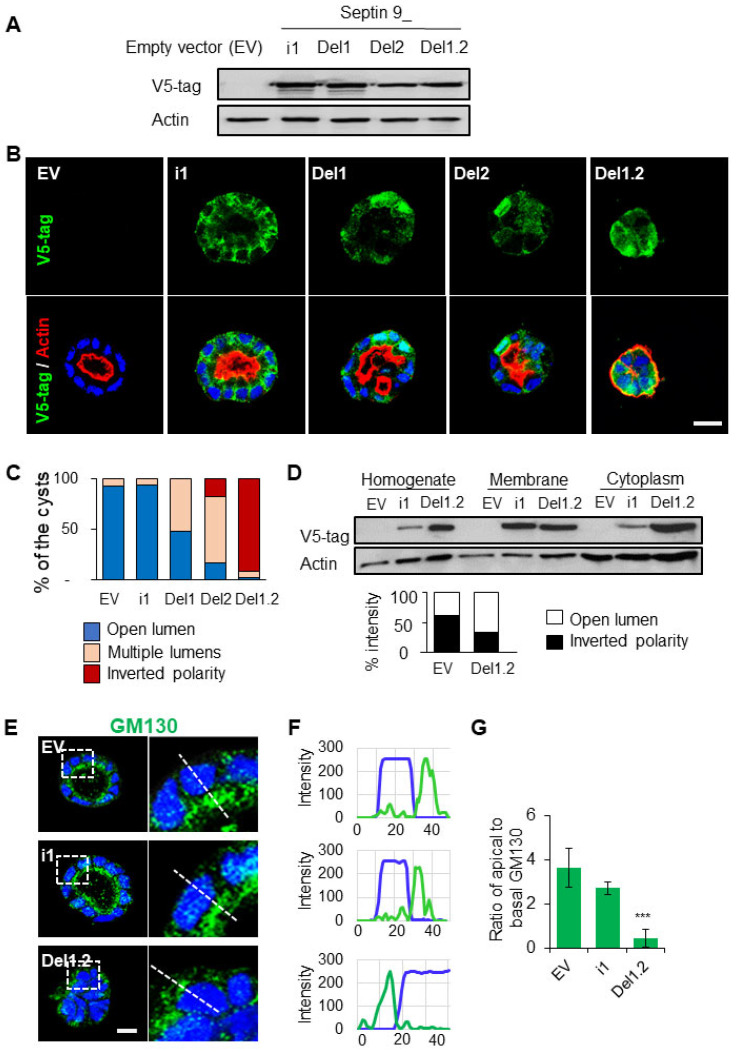
The deletion of septin 9 PB domains impacts different cyst phenotypes and septin 9 distribution in the cells. (**A**) The lysed MDCK stable cells transfected with septin 9_i1 (i1), septin 9_del1 (Del1), septin 9_del2 (Del2), and septin 9_del1.2 (Del1.2) or empty vector (EV) were tested via Western blot with septin 9−v5 tag antibody. (**B**) The MDCK stable cell lines were plated on Matrigel for 4 days to form cysts and then stained for septin 9−V5 tag (green) and actin (red). Septin 9_del1.2 (Del1.2) MDCK cell lines presented an inverted phenotype. A single confocal section through the middle of a cyst is shown. Scale bar 10 µm. (**C**) The color for each phenotype in a column corresponds to the color below the images. (**D**) MDCK stable cell lines were grown overnight before undergoing a subcellular fractionation assay and analyzed with Western blot for septin 9−v5 tag g and actin. The graph also shows septin 9−v5 tag densitometry analysis of the subcellular fractionation assay. (**E**) As during the experiment shown in (**B**), the cysts from MDCK stable cell lines of EV, i1, and Del1.2 were stained for Golgi (green). A single confocal section through the middle of a cyst is shown. Scale bar 10 µm. (**F**) Line profiles showing the Golgi distribution from the basal to the apical areas, obtained using Image J. (**G**) Quantification of Golgi distribution: the ratio of maximal fluorescence intensity at the apical side versus the maximal fluorescence intensity at the basal side. Data information: The data concern at least two replicates and cysts (*n* > 10) for 3D staining; three replicates for immunoblot. For D, *n* = 23 (EV), *n* = 29 (i1), *n* = 38 (Del1), *n* = 20 (Del2), *n* = 44 (Del1.2). The statistical values are means ± s.e.m. Student’s *t*-test was used. *** *p*  <  0.001.

**Figure 4 cells-12-01815-f004:**
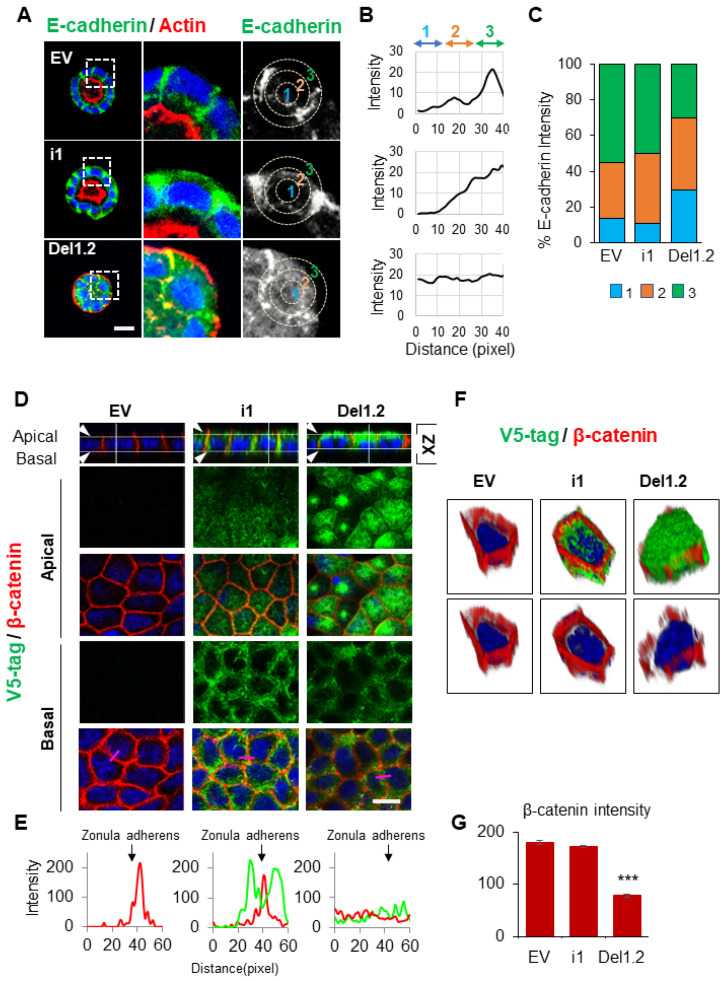
The deletion of septin 9 PB domains impacts cell–cell adhesion. (**A**) MDCK stable cell lines were plated on Matrigel for 4 days to form cysts and then stained for E-cadherin (green) and actin (red). A single confocal section through the middle of a cyst is shown. Scale bar 10 µm. (**B**) Distribution of the E-cadherin signal in cysts presented in (**A**) was analyzed using ImageJ software in an individual cell from a cyst. A circle was defined at the periphery of each cell, and the plugin produced a profile plot of normalized integrated intensities in concentric circles as a function of distance from a point in the image (considered here as the center of the cell). The circle was divided into three bands (1, 2, 3) with an equal radius. (**C**) The quantification of the fluorescence intensity of E-cadherin from each band is represented as a histogram. (**D**) MDCK stable cell lines of EV, i1, and Del1.2 were plated on Transwell filters for 24 h. Cells were fixed and stained for V5 tag (green) and β-catenin (red). Confocal images of XZ sections are presented at the top. A single confocal section of the apical and basal parts of the monolayer is shown in the middle. At the bottom of each column, the fluorescence of V5 tag (green) and β catenin (red) signals was scanned along the pink line drawn at a random position and presented in pink squares. Scale bar: 10 µm. (**E**) Arrowheads indicate the Zonula adherens. Bar graphs in red present the fluorescence intensity of β-catenin in the Zonula adherens. (**F**) 3D reconstruction of the septin 9 and β-catenin signals in a single cell from each group. (**G**) Bar graphs in red present the percentage of β-catenin intensity in the basolateral and apical zones. Data information: Data concern at least two replicates and cysts (*n* = 10) for 3D staining and cysts (*n* = 30) for 2D staining. The statistical values are means ±s.e.m. Student’s *t*-test was used. *** *p*  <  0.001.

**Figure 5 cells-12-01815-f005:**
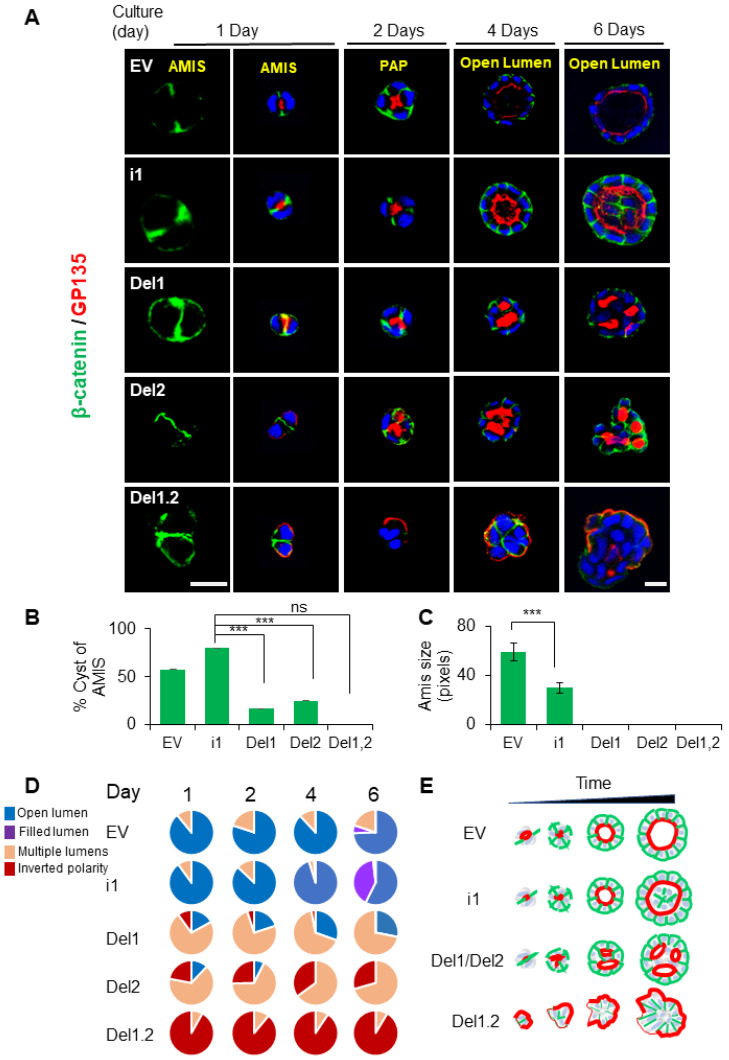
The deletion of septin 9 PB domains impacts cyst phenotypes from AMIS formation to lumen formation. (**A**) All MDCK stable cell lines were plated on Matrigel for 1 day, 2 days, 4 days, and 6 days to form cysts and then stained for β-catenin (green) and GP135 (red). A single confocal section through the middle of a cyst is shown. For the AMIS stage, β-catenin (green), the scale bar is 20 µm; the others are all 10 µm. (**B**) The analysis of AMIS formation of each stable cell line after 1 day is shown in the histogram. (**C**) Bars represent the size of the AMIS. (**D**) The phenotypes of open lumen, multi-lumen, inverted polarity, and filled lumen were counted during the culture periods. The color for each phenotype in the pie chart corresponds to the color below the images. (**E**) Schematic representation of septin 9 status and its effects on the phenotype of the MDCK cell cysts at the different stages. Data information: Data concern at least two replicates and cysts (*n* > 20) for 3D staining. The statistical values are means ± s.e.m. Student’s *t*-test was used. *** *p*  <  0.001.

**Figure 6 cells-12-01815-f006:**
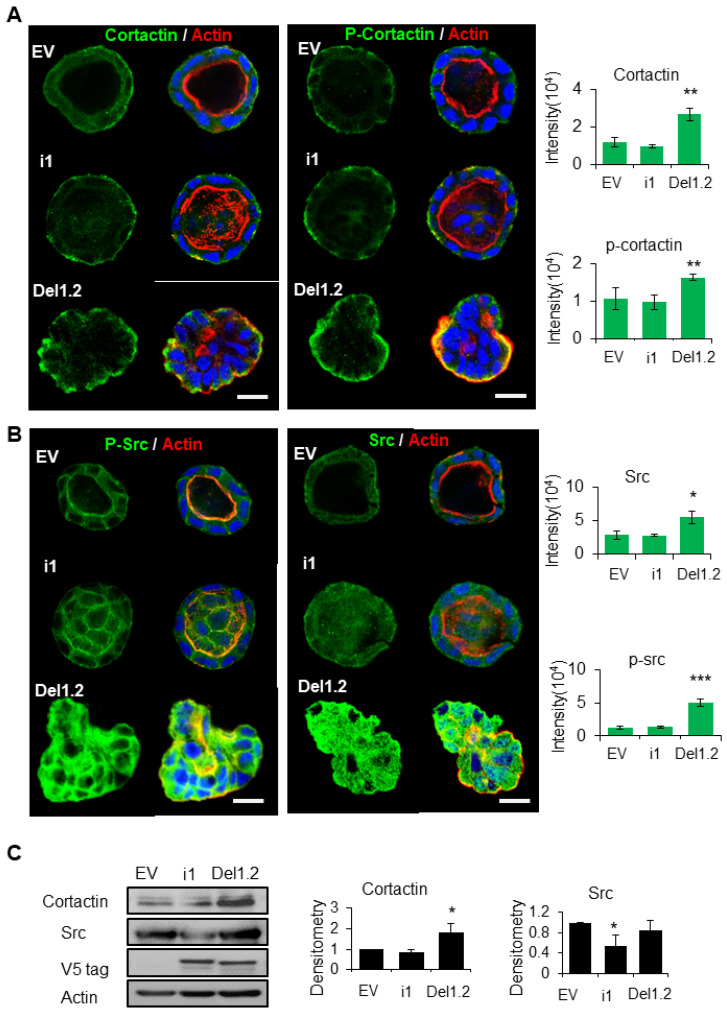
The deletion of septin 9 PB domains induces collective migration by regulating the src/cortactin signal pathway on day 6. (**A**,**B**) MDCK stable cell lines of EV, i1, and Del1.2 were plated on Matrigel for 6 days to form cysts and then stained for cortactin (green) and Actin (red) and p-cortactin (green) and Actin (red), src (green) and Actin (red), p-src (green) and Actin (red). A single confocal section through the middle of a cyst is shown. The scale bar is 10 µm. Quantification of the fluorescence intensity of each marker is shown beside the images. (**C**) Immunoblotting of the src and cortactin proteins and densitometry analysis. The data are means ±s.e.m. Student’s *t*-test was used. ** *p* < 0.01. Data information: Data concern at least two replicates and cysts (*n* = 10) for 3D staining and three replicates for immunoblotting. The statistical values are means ± s.e.m. Student’s *t*-test was used. * *p* < 0.05, ** *p* < 0.01, *** *p*  <  0.001.

**Figure 7 cells-12-01815-f007:**
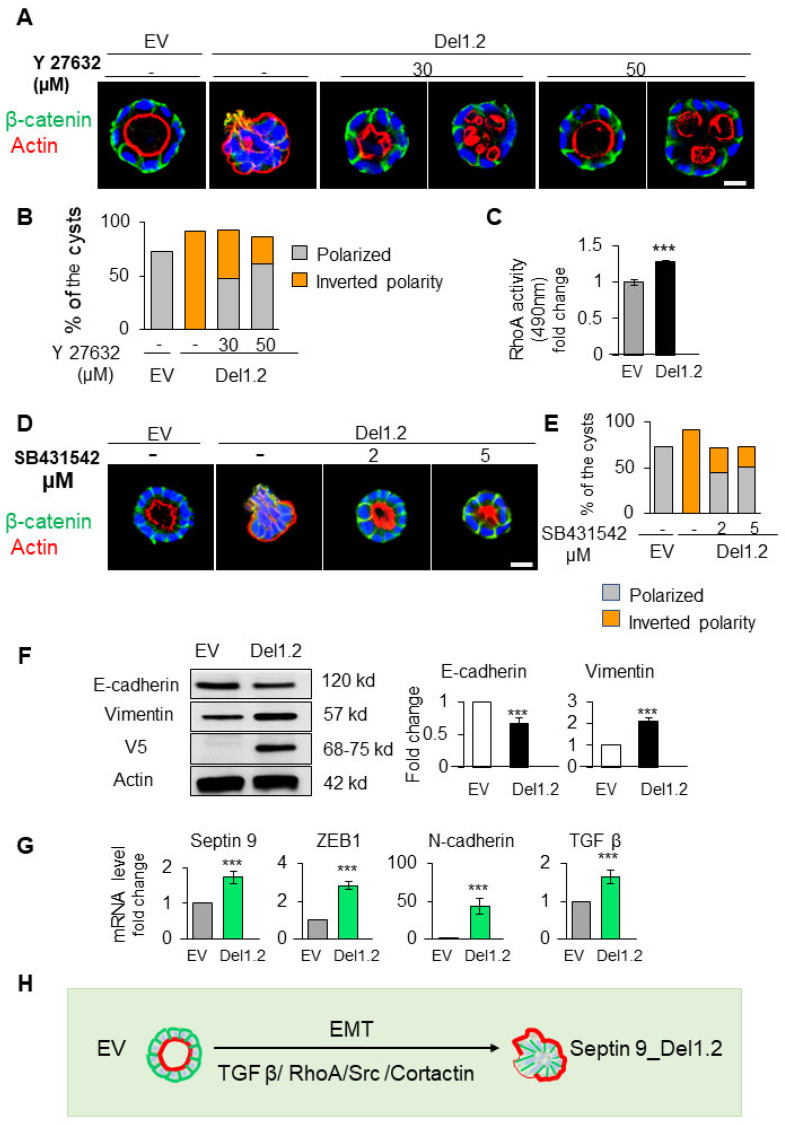
RhoA and TGF-β signals are involved in the inverted polarity phenotype of del1.2 cysts. (**A**) MDCK cells expressing EV and septin 9_del1.2 were plated on Matrigel for 6 days and treated with 30 µM and 50 µM of Y27632. All cysts were stained with β-catenin (green) for the basolateral membrane, actin (red) for the apical surface, and Hoechst (blue) for nuclei. The representative confocal images are shown merged (**B**) Quantification of polarized (lumen inside the cysts) and inverted polarity phenotypes in septin 9_EV and septin 9_del1.2 cysts. (**C**) Fold-change regarding RhoA activation in septin 9_EV and septin 9_del1.2 cysts. (**D**) MDCK cells expressing EV and septin 9_del1.2 were plated on Matrigel for 6 days and treated with 2 µM and 5 µM of SB431542. All cysts were stained with β-catenin (green) and actin (red). The representative confocal images for the merged signals are shown. (**E**) Quantification of cysts with a polarized phenotype and the inverted polarity phenotype is shown. (**F**) Western blot analysis of E-cadherin, vimentin, in cells with EV and septin 9_del1.2 cultured under 3D conditions. Actin is shown as a loading control. Quantifications of Western blot data for E-cadherin and vimentin protein levels. (**G**) qRT-PCR analysis of the expression of mRNA encoding septin 9, N-cadherin, ZEB1, and TGFβ in cells with EV and septin 9_del1.2 cultured under 2D conditions. (**H**) Schematic diagram illustrating the inverted polarity phenotype induced following the deletion of the two PB domains of septin 9_i1 in 3D culture. These plastic processes are EMT, and the process is regulated by the TGFβ/RhoA/Src/Cortactin pathway. Data information: Data concern at least two replicates and cysts (*n* > 10) for 3D staining and three for the RhoA activity test and immunoblotting and qRT-PCR. The statistical values are means ± s.e.m. Student’s *t*-test was used. *** *p*  <  0.001.

## Data Availability

The datasets supporting the current study have not been deposited in a public repository but are available upon request from the corresponding author.

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
