# Peer review of "Septin 9 Orients the Apico–Basal Polarity Axis and Controls Plasticity Signals"

_cells, 2023, doi:10.3390/cells12141815_

Round 1

Reviewer 1 Report

In this manuscript, Cai et al build upon previous research from the Gassama-Diagne lab investigating the role of Septin 9 in epithelial cells.  Septin 9 belongs to the large family of Septins that are proposed to hetero-oligomerize and potential form filaments that participate in diverse biological functions. Using both microscopy and biochemistry, Cai et al show Septin 9 plays a critical role in control apical-basal polarity and that the two polybasic domains in Septin 9 are essential for its normal function in this process.  Strikingly, loss of Septin 9 from MDCK cells results in an inversion of polarity, similar to the effect of overexpressing a variant of Septin 9 that lacks its two PB domains.  Interestingly, cysts with an inverted phenotype display an invasive phenotype that is proposed to be mediated via src and cortactin.

On the whole, this manuscript presents ample data to support the conclusion that Septin 9 plays a critical role in the establishment of apical-basal polarity in epithelial cells.  The experiments are well-done and sufficiently quantified to support their conclusions.  The text of the manuscript was often hard to read, with numerous grammatical mistakes, and therefore should receive editing before publication.

The authors should address the following minor points:

1)     The biggest issue is the apparent difference in localization of endogenous Septin 9 (Fig 1A) which appears be almost exclusively at the basal membrane, while V5-tagged Septin 9 i1 appears be mostly localize at the lateral membrane (Fig 3C, Fig 4D).  Do the authors have an explanation for this?

2)    Can the authors show the localization of Septin 9 in tissue samples (kidney preferably, since they are using MDCK cells).

3)    The authors should provide a domain diagram of the constructs used in this study (Sept 9 i1, Del1, Del2, Del1.2.  Furthermore, you should explain to the reader what “Sept 9 i1” is.  I assume it is the full-length Septin 9 isoform 1 construct.  This is not clearly communicated in the manuscript.

4)    The authors should also include the details of Sept 9 i1, Del1, Del2, Del1.2 in the methods, outlining specifically what sections of the protein are deleted.

5)    The experiments utilizing Septin 9 Del1, Septin 9 Del2, Septin 9 Del1.2 constructs appear to be done in the background of endogenous Septin 9 in these stable cell lines.  Do the authors believe that these constructs are acting as a dominant negative against endogenous Sept 9?  Would they have access to an available antibody that would allow them to visualize endogenous Septin 9 in the Septin Del1.2-expressing cell line (epitope directed towards the region deleted in Del1.2).

Needs improvement.

Reviewer 2 Report

The authors of the manuscript “Septin 9 orients apico-basal polarity axis and controls plasticity signals” have characterized the role of Septin 9 in regulating apico-basal polarity using 3D multure of MDCK cells as a model.  In figure 1 the authors demonstrated that siRNA knockdown of septin 9 results in MDCK cells cysts in 3D culture with multiple or inverted lumens, clearly demonstrating a key role for Septin 9 in apico-basal polarity in this system.  In figure 2 the authors demonstrate that knockdown of Septin 9 results in decreased expression of cell-cell adhesion proteins such as E-cadherin, N-cadherin, B-catenin and cortactin, and increases in expression of vimentin and ZEB1.  These expression changes are indicative of knockdown of septin 9 promoting an EMT.  In figure 3 the authors used deletion constructs to demonstrate that the polybasic (pb) domains were required for function in localization and role in apico-basal polarity.  In figure 4 the authors demonstrate that the pb deletion constructs disrupt cell-cell adhesion proteins such as E-cadherin and B-catenin.  In figure 5 they characterize the temporal role of septin 9 in lumen formation.  In figure 6 the authors show that expressing the deletion constructs induces cell behaviors similar to collective cell migration including phosphorylation of Src and Cortactin. In figure 7, they demonstrate that Septin 9 deletions signal through the RhoA pathway.   Overall, this manuscript is a solid characterization of the role of septin 9 in 3D cell culture and is a good fit for cells.  I recommend this be accepted with some minor corrections. 

Specific Comments:

Line 2.  The wording of the title is awkward.  Either say “apico-basal polarity and ...” or “the apico-basal polarity axis”

Line 10:  should read “ of the cell cortex”

Line 13:  should read “ localizes to the basolateral membrane.

Line 176: the phrase “was essentially presented at BM” is confusing and should be rewritten.

Figure 1 and later in the paper: it is a bit odd to write “siseptin”. The all lowercase formatting makes it more confusing for the reader than is needed. 

Line 224: The sentence “Loss of cortactin by septin 9 depletion was also validated in 3D culture by immunofluorescence analysis” is awkward and would be clearer to the reader if rewrite. 

Figure 3:  Starting with Figure 3 and continuing throughout the paper, the authors use deletion constructs in stable lines for most of their experiments.  They observed apico-basal defects in siRNA and in stable cell lines expressing Septin 9 lacking the PB domains.   It is important for the reader that the authors include a more detailed description of this experimental system.  Since these cells express Septin 9 and they are expressing the deletion construct, they should clarify if they believe they are observing a dominant negative or over-expression defect.  It might be worth testing if siRNA treatment worsens the defects found in the stable cell lines, which might be expected if the stable cell lines have dominant negative effects. 

Figure 7 legend:”  The title for this figure is the same as Figure 6, which appears to be a typographical error, since this figure is focused on RhoA not Src and Cortactin. 

Reviewer 3 Report

In this work by Ting Ting Cai et al entitled “Septin 9 orients apico-basal polarity axis and controls plasticity signals”, the authors aim to demonstrate the central role that septin 9 plays in regulating proper basolateral cell polarisation. Such changes in cellular polarisation is accompanied with differential expression of specific markers and resulted in differential cellular behaviours depending on what they studied. Further experiments were conducted when different version and truncated forms of the septin 9 protein were engineered and studies were carried out to see how they affected cellular localisation and/or expression of specific markers as well as overall changes in specific cell status. The work is of interest and delivers some important observations about the potential role of septin 9 in specific biological concepts but also offer some weaknesses throughout which reduce the impact of the delivery.   

Major points:

One of the key weaknesses of the work presented here is that only a single siRNA was used for the knocking down of septin 9. There is furthermore no clear indication as to the different controls used for the analysis, as in mock control, scrambled siRNA etc… which should be conducted for appropriate validation of the work carried out.

The parallel reduction is septin 7 and septin2 again raise the issues related to a possible cross-over and lack of specificity rather than the suggested feedback loop mechanism.

The original data for the different blots after septin9 siRNA treatment again show the presence of multiple bands in the vicinity of the one highlighted and it is not clear why the authors believe that the one they picked are the correct bands, raising further concerns about the specificity of the work in places.

Is septin 9 knockdown leading to cell death? Some of the date presented i.e. down-regulation of all markers looked at could be the result of cell apoptosis rather than any other specific processes.

How does the overexpression of the different v5-tag fused septin 9 compared to endogenous levels of the untagged septin 9. Furthermore, It is surprising to see that all deleted forms of septin 9 run at approximately the same size. Is this expected given the different truncations?

Have the authors analysed different clonal expansions expressing the different v5-tagged septin 9 and whether they see the same outcome suggesting that the differences seen are no clone specific?

In different places, the authors consider changes in protein expression as well as the degree of their activation/inactivation though phosphorylation but they never show quantitatively by Western blotting the latter whilst they do show changes in the overall expression of the former (i.e. cortactin/p-cortactin (Fig 6)).  It would be good for completeness that all the appropriate quantifications are conducted.

Numerous sections in the manuscript refers to the overall changes in cell migration or invasion but yet no real experiments were conducted to actually measure these parameters.

Minor points:

The introduction could provide further information about septin 9 as, as it stands, a single sentence related to this specific protein is given.

A figure of the different truncated septin 9 proteins (as further detailed from Fig 3A) and the different tags would help the reader comprehend.

The English is satisfactory but authors should consider that the introduction could provide further information about septin 9 as, as it stands, a single sentence related to this specific protein is given.

Author Response

The point by point responses are in the attached file

Round 2

Reviewer 3 Report

No further action needed